# Effects of a Digital, Person-Centered, Photo-Activity Intervention on the Social Interaction of Nursing Home Residents with Dementia, Their Informal Carers and Formal Carers: An Explorative Randomized Controlled Trial

**DOI:** 10.3390/bs15081008

**Published:** 2025-07-24

**Authors:** Josephine Rose Orejana Tan, Teake P. Ettema, Adriaan W. Hoogendoorn, Petra Boersma, Sietske A. M. Sikkes, Robbert J. J. Gobbens, Rose-Marie Dröes

**Affiliations:** 1Department of Psychiatry, Amsterdam Public Health Research Institute, Amsterdam University Medical Centers, Location Vrije Universiteit/De Boelelaan 1117, 1081 HV Amsterdam, The Netherlands; tp.ettema@icloud.com (T.P.E.); aw.hoogendoorn@amsterdamumc.nl (A.W.H.); rm.droes@amsterdamumc.nl (R.-M.D.); 2Faculty of Health, Sports and Social Work, Inholland University of Applied Sciences, Pina Bauschplein 4, 1095 PN Amsterdam, The Netherlands; petra.boersma@inholland.nl (P.B.); robbert.gobbens@inholland.nl (R.J.J.G.); 3Ben Sajet Centrum, Zwanenburgwal 206, 1011 JH Amsterdam, The Netherlands; 4Department of Clinical, Neuro- & Developmental Psychology, Faculty of Behavioural and Movement Sciences, Vrije Universiteit Amsterdam, De Boelelaan 1105, 1081 HV Amsterdam, The Netherlands; s.sikkes@amsterdamumc.nl; 5Neurology, Alzheimer Center Amsterdam, Vrije Universiteit Amsterdam, Amsterdam University Medical Centers, De Boelelaan 1105, 1081 HV Amsterdam, The Netherlands; 6Amsterdam Neuroscience, Neurodegeneration, De Boelelaan 1105, 1081 HV Amsterdam, The Netherlands; 7Kennemerhart, Diakenhuisweg 41, 2033 AP Haarlem, The Netherlands; 8Zonnehuisgroep Amstelland, Groenelaan 7, 1186 AA Amstelveen, The Netherlands; 9Department of Family Medicine and Population Health, Faculty of Medicine and Health Sciences, University of Antwerp, Universiteitsplein 1, 2610 Antwerp, Belgium; 10Tranzo, Tilburg University, Warandelaan 2, 5037 AB Tilburg, The Netherlands

**Keywords:** dementia, nursing homes, technology, photos, psychosocial intervention, artistic intervention

## Abstract

To enhance social interaction of residents living with dementia and their (in)formal carers in nursing homes, we examined the effects of a digital, person-centred, Photo-Activity (PA) versus a conversation activity (control). An explorative randomized controlled trial was conducted in 81 resident-informal carer (IC) dyads and 51 formal carers (FC) with three measurements (pre/post-test, 2-week follow-up). Intervention effects were tested using Mann–Whitney U’s, and ANCOVA’s with pre-test scores as covariates. Interaction effects were examined between dementia severity (DS; less/more) and condition (PA/control). A post-test effect was observed in social interaction (INTERACT-subscale: Mood [*p* = 0.037, *ηp*^2^ = 0.07]), with PA residents showing better mood than controls. Residents with less DS showed more positive effects of PA than residents with more DS (interaction effects: INTERACT-subscales Mood [*p* = 0.017, *ηp*^2^ = 0.092], Stimulation Level [*p* = 0.011, *ηp*^2^ = 0.106], and Need for Prompting [*p* = 0.013, *ηp*^2^ = 0.099]). Higher QUALIDEM Positive Affect scores were observed in the PA group, post-test (*p* = 0.025, *ηp*^2^ = 0.082), and follow-up (*p* = 0.042, *d* = 0.39). PA FC showed less empathy (IRI; *p* = 0.006, *ηp*^2^ = 0.185;) than controls, but reported getting to know the residents better (*p* = 0.035, *r* = 0.299). PA improved mood and positive affect of residents with dementia and led to FC knowing the residents better. Less empathy was observed in FC providing PA, requiring further investigation.

## 1. Introduction

For people living with dementia, being acknowledged for who they are as a person with unique needs and wishes, rather than being identified solely by their disease, is an important aspect of their social health ([26]). Being able to participate in meaningful social activities is one of the domains of social health highlighted by the INTERDEM Social Health Taskforce ([28]). Social participation maintains personhood for the person with dementia both living in the community and in long term care settings. Interventions that are effective in improving social participation of people with dementia often have these characteristics in common: they facilitate communication between the person with dementia and the people around them, they are attuned and tailored according to the person’s unique identity, and aim to provide positive and meaningful experiences rather than being task-focused ([28]; [50]; [51]).

Opportunities for social participation are often lacking for nursing home residents with dementia, with half of their time being unoccupied or spent alone ([45]; [60]). Previous studies found that nursing home residents can spend up to 17 h a day lying awake in bed, with feelings of boredom prevailing, in daily routines that revolve mostly around meal times ([11]; [34]). This is concerning because if residents are not involved in social activities in the nursing home, they are at risk of having their mood and social health negatively affected, along with their overall quality of life ([45]; [68]; [71]).

People that could provide necessary social interactions, like the formal and informal carers, may struggle to know how best to interact with the resident with dementia. For formal carers, being overwhelmed with other care tasks could also be a barrier to meaningful social interaction, because there is little opportunity for them to get to know the residents as individuals ([56]; [70]). In general, the way that formal carers perceive and approach residents with dementia can have an impact on resident outcomes via the quality of care ([6]). Fostering empathy has beneficial effects on the well-being of both formal carer and resident ([17]). Together with empathy, having a positive approach towards dementia as a formal carer has been described as a key component in delivering excellent care ([16]; [40]).

For informal carers, social interaction with the resident could be impeded when visits become uncomfortable due to how the relationship has changed with the progression of the dementia, where there are less shared experiences and understanding in the dyadic relationship ([14]; [62]). In terms of their interactions with the formal carers, family and informal carers commonly feel that their knowledge of the resident is insufficiently utilized, and their role in the resident’s new life in the nursing home is diminished ([3]). Providing more opportunities for informal carers to communicate and share their knowledge of the resident as a person could help informal carers feel recognized and increase their sense of competence ([15]; [33]; [66]).

Social engagement could be increased through interventions designed to enhance the social interaction between nursing home residents with dementia and their formal and informal carers ([60]). Adapting activities to the current skills and cognitive functioning of the residents makes it less likely that they lose interest in the activity ([18]; [70]; [37]). This was shown in a study where a reminiscence activity resulted in the residents with dementia speaking more and sharing stories that were more detailed and had personal significance when viewing generic photos versus personal (i.e., family) photos ([5]).

Activities also need to be person-centred and relate to the person in a meaningful way, otherwise the residents are less likely to participate in the activity ([70]; [1]). Individually tailored activities are found to be more effective in terms of reducing agitation and increasing positive affect ([54]; [71]). For example, in a group of men with dementia, positive outcomes like enjoyment and active involvement in the group were observed during a group activity that revolved around discussing digitalized photos of national Scottish football, which was a common interest among the participants ([78]). A randomized controlled pilot Photo-Activity study, using generic photos in conversations showed more positive tendencies for speech, mood, negative behaviour and social interaction in the group of residents with dementia that used person-oriented photos (i.e., the photos matched the personal interests of the resident), compared to the group of residents that used non-person-oriented photos ([77]). As the study sample was too small (n = 20) to detect statistical significant differences, it is worth exploring this person-centred Photo-Activity further as it highlighted the potential benefits of using generic photos tailored to the interests of the person with dementia. Moreover, it would be relevant to also examine the effects of the Photo-Activity on formal and informal carers. A systematic review, conducted by [73] ([73]) found that current studies on psychosocial interventions that use generic photos are limited and have varying methodological quality.

To address these limitations and gaps, we conducted an explorative randomized controlled trial (RCT) investigating the effects of a digital person-centred Photo-Activity versus a control activity (general conversation) on nursing home residents with dementia and their (in)formal carers. Based on the findings of the pilot study ([77]) and previous literature on the use of generic photos ([5]), it was expected that residents who did the Photo-Activity with their carers would feel more seen and acknowledged as a person, and show improved social interaction, mood, and quality of life. By providing opportunities for knowledge exchange between the informal and formal carers about the interests of the resident with dementia and the residents’ reaction to the Photo-Activity, and by involving informal carers in offering the Photo-Activity if they wanted to, it was expected that informal carers would develop a better sense of competence, and feel more recognized in their role as informal carers. In addition, it was expected that engaging in the Photo-Activity with residents would improve formal carers’ person-centred attitude, increase empathy, and would also allow them to know the residents and their family better. Finally, because of the above-mentioned expected impacts of the intervention on the three parties (resident, informal and formal carers) it was hypothesized that the Photo-Activity intervention would strengthen the relationship between people with dementia, informal carers and formal carers.

Based on the expectations mentioned above, the research questions central in our study and the corresponding hypotheses were:

Compared to the residents in the control activity, do residents in the Photo-Activity benefit more from the intervention regarding social interaction, behaviour, mood, feeling known, feeling satisfied with their stay in the nursing home, and quality of life?

**Hypothesis 1.** *Residents in the Photo-Activity will show better outcomes in social interaction, behaviour, mood, feeling known, feeling satisfied with their stay in the nursing home, and quality of life at posttest (after 4 weeks intervention) and at follow-up (two weeks after ending intervention), compared to residents in the control activity, taking into account their baseline status.* Compared to the informal carers in the control activity, do informal carers in the Photo-Activity have better sense of competence, feel more recognized by nursing home staff in their role as informal carer, feel more satisfied with the stay of their family member in the nursing home, and feel that their family member is more known as a person in the nursing home?

**Hypothesis 2.** 
*Informal carers in the Photo-Activity will have better sense of competence, feel more recognized by nursing home staff in their role as informal carer, feel more satisfied with the stay of their family member in the nursing home, and feel that their family member is more known as a person in the nursing home at posttest (after 4 weeks intervention) and follow-up (two weeks after ending intervention), compared to the informal carers in the control group, taking into account their baseline status.*


Compared to formal carers in the control activity, do formal carers in the Photo-Activity develop a more person-centred attitude, have more empathy, and know the person with dementia and their informal carer better as a result of the activity?

**Hypothesis 3.** 
*Formal carers in the Photo-Activity will show a more person-centered attitude, have significantly more empathy and know the person with dementia and their informal carer better as a result of the activity at posttest (after 4 weeks intervention), compared to the formal carers in the control activity, taking into account their baseline status.*


## 2. Materials and Methods

### 2.1. Design and Randomization Procedure

An exploratory RCT was conducted to answer the research questions. The CONSORT 2025 and TIDieR checklist was used to guide the reporting in this paper ([47]; [46]; Appendix A). Exploratory RCT’s often involve smaller sample sizes and shorter intervention periods compared to traditional RCT’s ([32]). Residents with dementia, together with their (in)formal carers, were recruited in matched pairs based on the severity of dementia ([65]; see Section 2.2 Setting and Participants), then randomized by the researchers through drawing lots to an experimental group who did the digital Photo-Activity intervention, versus a control group who did a general conversation activity for a period of four weeks. Formal carers were also randomized into the Photo-activity or control condition by drawing lots. In the first and last week of the intervention period data were collected on social interaction and mood of the person with dementia during the intervention sessions. Outcomes regarding the person with dementia’s social interaction, behaviour and mood in daily life, and quality of life, the informal carer’s sense of competence and the formal carer’s empathy and attitude towards dementia were measured at three time points: at baseline (T0), after the four-week intervention period (T1), and two-weeks later (T2), using quantitative methods. The full protocol for the exploratory RCT and the process evaluation that was performed alongside it, has been previously published ([72], [74]).

### 2.2. Setting and Participants

This research was assessed on 29 April 2020 by the Medical Ethics Review Committee of the VU University Medical Center, and was confirmed as not needing official approval of the committee, because the Medical Research Involving Human Subjects Act (WMO) does not apply. It was pre-registered in the Overview of Medical Research in the Netherlands (OMON, previously known as the Dutch National Trial Register; NL9219; https://www.onderzoekmetmensen.nl/nl/trial/25910 (accessed on 21 January 2021)).

The study was conducted in 20 participating nursing home wards across three healthcare organizations in the Netherlands. Data collection began in March 2021 and ended in January 2023. Based on the pilot study of the Photo-Activity ([77]), where large positive effect sizes were found for the INTERACT sub-scales ([10]; [13]) reduced negative behaviour (d = 0.85) and social-interaction (d = 0.86), a power calculation based on [19] ([19]) showed that for the exploratory RCT a sample size of 45 residents in both experimental and control condition was needed (power = 0.80, α = 0.05, d = 0.8), including an expected drop-out rate of 10% in the intervention period and follow-up (6 weeks in total).

Participants’ informed consent was collected by the study Coordinator, once participants were deemed to meet inclusion criteria. All participants in the study (person with dementia, their informal carer, and formal carer) gave their informed consent. In some cases where the person with dementia is unable to give their informed consent and sign the form, their informal carer who was also their legal representative, signed the form for them.

Inclusion criteria for the residents with dementia were: having a Global Deterioration Scale (GDS; [65]) score of 4 (moderate cognitive decline), 5 (moderately severe cognitive decline) or 6 (severe cognitive decline) based on assessment of one of the ward staff members assigned the role of Coordinator in the study (often the ward manager; served as the first point of contact for the resident and the family/informal carers); and living in the nursing home for at least one month. Residents that had severe vision or hearing problems according to the ward staff, were excluded from participation.

Together with the managers, three to four formal carers per ward with the following roles were invited to join the study: the Coordinator, the Independent Assessor (often combined with the Coordinator role, this involved observing how participating residents functioned in the ward on a daily basis during the intervention period and filling in observation scales at three time points, while being blinded to the condition assigned to the resident), the Photo-Activity carer (delivered the Photo-Activity to the resident), and the Control activity carer (delivered the general conversation activity to the resident). The two formal carers who agreed to take on one of the activity roles were randomized into either the experimental or control condition. Participating formal carers were urged not to discuss the activities they were assigned to, in order to prevent influencing each other.

Informal carers of the included residents were also invited by the researchers to deliver either one of the activities to their family member residing in the nursing home. The informal carers were still involved as participants whether or not they chose to deliver either of the activities.

### 2.3. Interventions

#### 2.3.1. Digital Photo-Activity (Experimental)

The Fotoscope web-app, used in the digital Photo-Activity, has been developed by a multi-disciplinary team with various backgrounds of expertise and lived experiences (i.e., researchers in dementia, visual artists, web designers, software engineers, as well as residents with dementia and their (in)formal carers). It contains a database of 1500 black and white artistic photos, curated by the visual artist Laurence Aëgerter, building from her earlier work in 2016 called the *Photographic Treatment* ([2]). The photos are categorized according to the main seven themes of People, Places, Nature, Animals, Things, Activities, and Experiences, and sub and sub-sub categories to make it easy for the user to find a specific photo. The Fotoscope web-app is divided into five main pages: Themes, Profile, Favorites, User Guide, and Information ([74]). Through the Fotoscope, carers and residents with dementia could view and talk about beautiful black and white, high-quality, digital photos that relate to the personal interests of the resident. Because the photos are generic (not photos from the residents’ own life), it is intended that knowledge questions from carers are minimized and the residents will feel more comfortable in a conversation that gives them more opportunity for talking about their feelings, insights and opinions, rather than what they do or do not remember ([5]).

#### 2.3.2. General Conversation Activity (Control)

The carers in the control condition were instructed to have an open conversation about general topics together with the resident, in the absence of the tablet and the (digital) photos. The general conversation activity was chosen as an active control intervention because it closely resembles usual care, and because we wanted to eliminate the possibility of seeing positive effects from the Photo-Activity that may have been due to the residents receiving more individual attention from the formal carers.

### 2.4. Outcome Measures and Procedure of Data Collection

Residents, formal and informal carers completed questionnaires during the different intervention time points (Table 1; [72]). Participants’ background characteristics were collected via selected questions from the TOPICS-MDS ([59]). Informal carers provided information regarding background characteristics of the residents.

#### 2.4.1. Primary Outcomes

##### Residents

The resident’s social interaction with the carer during the Photo-Activity or the general conversation activity was measured in the first and last intervention session using the INTERACT ([10]) observation scale (in English) by trained student researchers who observed the session online by video call. The INTERACT ([10]) was originally developed to measure relevant domains that were thought to be affected by a Multi-Sensory Stimulation intervention for people with dementia, like mood and speech ([80]). It was previously found to have a test–retest reliability of 0.99 (Pearson correlation coefficient) and was also found to have a medium to high inter-observer reliability ([9]; [12]). In this study, an adapted 34-item INTERACT, from a previous study of an art-based (theatre) activity in the nursing home, was used ([13], [12]).

Quality of life (QoL) of the resident with dementia was assessed using the Dutch 37-item QUALIDEM ([30]) by the Independent Assessor, based on their observations of the resident one week prior T0, T1 and T2. The QUALIDEM, which was found to have good reliability and validity was designed to measure quality of life of residents with dementia in the nursing home, from the view of the formal carer, and was developed in the context of Dutch nursing homes.

#### 2.4.2. Secondary Outcomes

##### Residents

The Smiley Face Assessment Scale (SFAS; [41]; [67]) was used to measure self-reported mood of the resident immediately before and after the activity sessions in the first and last week of the intervention period. The resident was asked by the carer how they were feeling in the moment, and the resident could answer by picking one of five faces depicting moods from ‘very unhappy to very happy’. While no validity studies have been performed for the SFAS specifically, it has been used previously in other studies that also involved people with dementia ([41]; [67]). Good validity however has been found for similar visual analogue scales that measure mood in people with dementia ([58]; [76]). In the process evaluation that was conducted within this RCT, only a minority of formal carers reported that the residents had difficulty understanding the SFAS ([74]).

The Dutch version of the brief Neuropsychiatric Inventory Questionnaire (NPI-Q, 10-item; [49]), which was found to have concurrent validity ([24]), acceptable test–retest reliability (with Pearson correlation between 0.80 and 0.94; [49]), and good internal consistency ([38]) was completed by the Independent Assessor at T0, T1 and T2, to measure the presence and severity of residents’ behavioural symptoms in the past week, including delusions, hallucinations, agitation, depression, anxiety, euphoria, apathy, disinhibited behaviour, irritability, purposeless repetitive behaviour ([49]).

Questions regarding feeling known, specifically composed for this study, were asked of the residents in the experimental and control group. The residents were asked by the Independent Assessor at the end of the intervention period to what extent they felt more known as a person by their formal carer, and how satisfied they were with their stay in the nursing home, on a 10-point Likert scale, with 1 being the lowest and 10 being the highest.

##### Informal Carers

The informal carers were asked to complete the Short Sense of Competence Questionnaire (SSCQ; [84]) at T0, T1 and T2. This, originally Dutch, scale has 7 items around possible problems that the informal carer might experience in caring for the resident and indicates the degree to which the carer feels capable of caring for the person with dementia. The SSCQ was found to be valid and reliable in previous research among the target group ([84]). The SSCQ has been widely used for Dutch informal carers of people with dementia ([63]; [81]).

Informal carers also answered questions about how much they felt that the nursing home staff recognized their role as informal carer of the resident, how satisfied they were with the care in the nursing home and to what extent the informal carers felt that their loved one was recognized as a person in the nursing home. These questions could be answered by choosing a number on a 10-point Likert scale, with 1 being the lowest, and 10 the highest.

##### Formal Carers

Person-centred attitude towards dementia and empathy towards the residents of formal carers at T0 and T1 were measured using the Approaches to Dementia Questionnaire (ADQ; [57]) and the Interpersonal Reactivity Index (IRI; [22]), respectively. It was decided at the start of the trial by the project group, in consultation with the formal carers recruited in the study to limit data collection to T1 (so no follow-up) to reduce the burden on formal carers. The ADQ has 19 items covering two domains- Hope and Person-centeredness. It was found to be valid and have good test–retest reliability and is often used in long-term care settings ([25]; [57]). The ADQ was translated to Dutch by previous researchers and validated professionally by an English translator ([36]).

The IRI has 28 items, measuring global empathy among formal carers, and was also found to be valid and reliable ([23]). The Dutch version of the IRI was translated following back-translation procedure and was found to have satisfactory internal consistency.

Formal carers in the experimental and control group were asked at T1 how much they got to know the resident and the informal carer through the activity: “To what extent did you get to know the person with dementia and their relative/loved one better through the activity?”. Formal carers could also choose a number on a 10-point Likert scale, with 1 being the lowest (‘did not get to know the resident and informal carer at all’) to 10 being the highest (‘got to know the resident and informal carer much better’). In the process evaluation, formal carers were asked if they got to know the person with dementia (only) better through the activity (answer options of ‘Yes, much better’, ‘Yes, a little better’, ‘No, not better’).

### 2.5. Procedure

As the COVID-19 pandemic severely limited access and visits to nursing homes ([35]), the data collection protocol of the RCT was partly adapted into an online protocol. The researchers trained the formal carers online for their different roles and observed online while the residents with dementia and their formal carers did the activities together in the nursing home.

Formal carers assigned to the role of Coordinator were trained to assess the dementia severity of the residents using the GDS ([65]). Formal carers assigned to be Independent Assessor were trained on how to observe the residents using the QUALIDEM ([30]) and NPI-Q ([49]). Both trainings were done by a senior researcher for about 30 min to one hour.

Formal carers assigned to the Photo-Activity received a 1.5 h online training delivered by a senior and junior researcher before providing the intervention to residents. In the training, the researchers discussed general information about the research, the background of the Photo-Activity and the procedure for engaging in the activity with the residents, provided an online demonstration of how to select photos in the Fotoscope app, and gave general and person-centred communication tips. Carers were given their own personal Fotoscope profile with log-in credentials so they could practice using the web-app whenever they liked.

Formal carers in the control condition received a one-hour online training where they received brief information about the research, instructions for the general conversation activity, and general communication tips.

The Photo-Activity was delivered by the formal carer in two 30 min sessions a week for four weeks via the Fotoscope (Figure 1; [72]). The formal carers were advised to first prepare for the Photo-Activity by interviewing either the resident or a family member about the personal interests of the person with dementia using the questions on the Profile page of the Fotoscope, and then using the selection tool on the app to select and save at least 15 photos on the Photo Selection page which reflected the residents’ personal interests. Once the photos were prepared, carers were instructed to schedule two sessions per week of about 30 min per session, for four weeks, where they could sit down with the resident in a quiet room, offer a drink and view the photos together. Formal carers were also asked to get in touch with the informal carer at least once or twice during the four-week intervention period, by giving them updates about how the activity with the resident was going, either by email or telephone call, or whenever the informal carer came to visit.

The carers in the control activity were asked to deliver the activity with the same frequency and for the same period (twice a week, 30 min per session, for four weeks). They did not receive any instructions to update the informal carers.

Formal carers were asked to set-up a video-call with a laptop or tablet before starting the first and last intervention session so that trained research assistants could join in and observe the interaction between carer and resident during the session using the INTERACT ([10]). Student researchers were trained to introduce themselves to both carer and resident at the start of the session, to ensure that the camera and microphones were working correctly, and to remind the carer to administer the SFAS ([41]) before and after the session. Based on comments from the first participants, researchers turned off their cameras after their introduction and put themselves on mute during the call so as not to distract the residents. For the experimental group, immediately after ending the session, and bringing the resident back to the living room, the research assistants gave feedback on how the carer could improve on their person-centred communication with the resident during the Photo-Activity. For the control group, the student researchers thanked the carer and resident immediately after the session and ended the call.

As part of the process evaluation that ran parallel to the RCT, residents, informal carers and formal carers were asked to complete semi-structured interviews after the final intervention and control session took place ([74]).

### 2.6. Statistical Analysis

Data was analyzed using IBM^®^ SPSS^®^ 28 for Windows. Background characteristics of the residents, informal and formal carers in the Photo-Activity and general conversation activity were analyzed with descriptive statistics and tested for between-group differences at baseline with *t*-test, Chi-square test, or Mann–Whitney U test, depending on the type of data.

First, to answer the three research questions, a one-way Analysis of Covariance (ANCOVA) was performed for T1 while including the baseline scores (T0) as covariate in the analysis for all data. Second, to explore for possible differences between sub-groups, a two-way ANCOVA was performed to test the interaction effects between-group allotment (experimental and control), and two classification factors, i.e., dementia severity (less or more dementia severity), and hours of visit of the informal carer to the resident in the past week before baseline measurements were taken (low or high hours; asked of the informal carer during collection of background characteristics at baseline) on the outcomes at T1. Finally, mixed model for repeated measures (MMRM) was performed for data that was collected over three time points (QUALIDEM; [30]), NPI-Q ([49]), SSCQ ([84]), feeling known questions for resident and informal carers) to see the overall effects of the intervention over time (T0, T1, T2), as well as the interaction effects between the intervention and the classification factors, at all time points. Intention-to-treat and ‘per protocol (completers) analyses’ were performed.

For self-reported mood of the residents around the activity sessions (SFAS; [41]) it was tested if participants in the experimental group had a more positive mood at the end of the activity sessions compared to before the sessions than participants in the control group. All scores of the before-session SFAS ([41]; pre-test) in the first and last week of the intervention period were therefore added up and compared with total scores of the after-session (post-test) SFAS ([41]). Between-group differences were tested with an ANCOVA on post-session scores, including pre-session scores as covariate in the statistical analysis.

For the question asked at T1 to the formal carers regarding whether they got to know the resident and the family better through the activity, Mann–Whitney U tests were performed to see if there were any differences in mean scores between the experimental and control group.

## 3. Results

A total of 81 residents were recruited at T0 (baseline). A total of 19 residents dropped out during the intervention period (Figure 2), leaving 62 residents at T1 (post-test). Additionally, 81 informal carers were recruited at T0, while 60 informal carers remained at T1. Finally, 51 formal carers were recruited at T0, while 41 remained at T1. Recruitment ran from March 2021 and ended in January 2023. Information on how the intervention and control activity were delivered and experienced by participants can be found in the process evaluation published elsewhere ([74]).

No differences in background characteristics between the experimental and control group were found (Table 2). Despite efforts from the research team and nursing home managers, there were no informal carers who wanted to participate in the study to deliver either of the activities to the residents for one month (most common reason was lack of time). Student interns from clinical psychology were trained and also participated in delivering the Photo-Activity (to 3 residents) or control activity (to 3 residents) during the data collection period, due to staffing issues in the nursing home. No data from the student researchers themselves were collected, but data from the residents whom they interacted with, and their informal carers were collected.

### 3.1. Primary Outcomes

**Residents (Research Question 1).** 
*Outcomes of the intervention regarding social interaction, behaviour, mood, feeling known, satisfied with the nursing home, and quality of life.*


*Social interaction.* A significant positive, medium effect (*F*[1, 61] = 4.56, *p* = 0.037, *partial eta*^2^ = 0.07) was found for the INTERACT ([10]) subscale Mood, where residents in the experimental activity had a more positive mood at T1, compared to the mood of the residents in the control activity (Table 3a). No significant main effects were found for the other INTERACT subscales. However, of note is the INTERACT sub-scale Negative Interactions, which almost reached statistical significance (*F*[1, 61] = 3.88, *p* = 0.053, *partial eta*^2^ = 0.060), where it was observed that residents in the experimental condition exhibited fewer negative interactions at T1, compared to residents in the control condition.

Cronbach’s alpha was calculated for the six out of seven sub-scales of the INTERACT (the Need for Prompting sub-scale only had one item and was excluded): Cronbach’s alpha coefficient for Mood-subscale was 0.83, for Speech was 0.82, for Relating to person was 0.50, for Relating to Environment was 0.38, for Stimulation level was 0.75, and for Negative Interactions was 0.50.

When dementia severity (DS) was included in the analyses, significant interaction effects of the group (experimental versus control condition) and DS (less dementia severity versus more dementia severity) were found for the INTERACT ([10]) subscales Mood (*F*[1, 59] = 6.00, *p* = 0.017, *partial eta*^2^ = 0.092), Need for Prompting (*F*[1, 59] = 6.51, *p* = 0.013, *partial eta*^2^ = 0.099), and Stimulation level (*F*[1, 59] = 6.97, *p* = 0.011, *partial eta*^2^ = 0.106; Table 4). Residents with low DS in the experimental activity had a higher score on the mood, stimulation level, and need for prompting (they needed less prompting) subscales compared to residents with low DS in the control group. Meanwhile the residents in the experimental group with high DS had lower scores on mood, stimulation level and need for prompting subscales, compared to the residents with high DS in the control group. No significant interaction effects were found between the group of residents (experimental versus control) and informal carers’ hours of visit.

*Quality of life.* The Positive Affect subscale of QUALIDEM ([30]) showed a significant difference between the groups, where the experimental group had a significantly higher score in Positive affect, i.e., better positive affect, compared to the control group at T1 (Table 3a), with medium effect size (*F*[1, 59] = 5.29, *p* = 0.025, *partial eta*^2^ = 0.082). No significant interaction effects were observed between group and dementia severity (Appendix A) or hours visit of the informal carer (Appendix A).

Cronbach’s alpha was found to be between 0.59 and 0.89 for the different subscales in a previous study ([31], [30]). In the current study, Cronbach’s alpha coefficient ranged from 0.25 to 0.89. Specifically, per sub-scale, Cronbach’s alpha coefficient was 0.81 for Care relationship, 0.89 for Positive affect, 0.78 for Negative affect, 0.79 for Restless tense behaviour, 0.55 for Positive self-esteem, 0.38 for Social relationships, 0.50 for Social isolation, 0.81 for Feeling at home, and 0.25 for Have something to do.

The results of the MMRM for QUALIDEM (Table 5) showed the same results as the ANCOVA, where the experimental group had a significantly higher score at T1 in the sub-scale of Positive Affect, with small effect size (*p* = 0.028, *d* = 0.38). This effect was also seen at T2, where the experimental group also had a significantly higher score compared to the control group (*p* = 0.042, *d* = 0.39), with a small effect size.

There were no significant interaction effects between group and dementia severity or hours visits by the informal carer at T1 or T2.

No other significant differences in main effects (Table 3a) or interaction effects (Appendix A) between experimental and control group were found for behaviour, mood, or questions about feeling known outcomes. The MMRM also showed no other significant main (Table 5) or interaction effects (Appendix A) between the experimental and control groups.

### 3.2. Secondary Outcomes

**Informal Carers (Research Question 2).** 
*Outcomes of the intervention on informal carers’ sense of competence, feeling recognized by nursing home staff in their role as informal carer, feeling satisfied with the stay of their family member in the nursing home, and feeling that their family member is known as a person in the nursing home.*


No significant differences in sense of competence (SSCQ; [84]) or questions to the informal carers regarding feeling recognized or known were found between the experimental and control groups (Table 3b). Also, no significant interaction effects were found between group and dementia severity or hours of visit of the informal carer (Appendix A).

The results of the MMRM for the SSCQ ([84]) and feeling recognized/known questions did not show any significant differences between the experimental and control group at T1 and T2 (Table 5). No significant interaction effects were found between group and dementia severity, or group and hours of visit of the informal carers at T1 and T2 (Appendix A).

A previous study had found that the SSCQ had a Cronbach’s alpha of 0.76 ([85]). For our sample, we found Cronbach’s alpha coefficient to be 0.88.

**Formal Carers (Research Question 3).** 
*Outcomes of the intervention on formal carers’ per-son-centred attitude, empathy, and knowing the person with dementia and their informal carer better as a result of the activity?*


*Empathy*. A significant difference with a large effect size was found in global empathy (IRI; [21]) between the experimental and control group (*F*[1, 38] = 8.65, *p* = 0.006, *partial eta*^2^ = 0.185). The control group scored higher in global empathy compared to the experimental group at T1 (Table 3c). Looking at the four subscales of the IRI ([21]) (fantasy, perspective taking, empathic concern, personal distress), it was found that the experimental group scored significantly lower on the sub-scales fantasy (*F*[1, 38] = 9.70, *p* = 0.003, *partial eta*^2^ = 0.203) and empathic concern (*F*[1, 38] = 7.62, *p* = 0.009, *partial eta*^2^ = 0.167), with large effect sizes (Table 3c). Cronbach’s alpha for the IRI subscales was previously found to be between 0.73 and 0.83 ([23]). In this study, Cronbach’s alpha coefficient was found to be 0.67 for the Perspective taking subscale, 0.64 for the Empathic concern subscale, 0.77 for the Fantasy subscale, and 0.67 for the Personal distress subscale.

*Approaches to Dementia.* To check whether baseline scores on approaches to dementia had an effect on formal carer’s empathy at T1, total ADQ ([25]) at baseline was divided into two classification categories (low baseline score versus high baseline score, where the median was used to determine the cut-off point), and was entered into the analysis as a second independent variable. A significant interaction effect with large effect size (*F*[1, 36] = 4.95, *p* = 0.032, *partial eta*^2^ = 0.121) was found. The experimental group scored lower in global empathy at T1 com-pared to the control group, regardless of whether they scored low or high for baseline Total ADQ ([25]), but larger differences between experimental and control were observed if total ADQ ([25]) at baseline was high, i.e., if carers scored high on person-centred attitude and hope (Table 6). No significant interaction effects were found between global empathy of the formal carer and resident’s dementia severity (Appendix A). There was a significant difference found in terms of how much formal carers in the experimental group got to know the resident with dementia (only) compared to the control group (*p* = 0.035, r = 0.299; Table 3c), with medium effect size. Formal carers in the experimental group reported getting to know the person with dementia ‘a little better’ (47.8%) or ‘much better’ (39.1%), compared to the control group (63.0% and 14.8%, respectively).

No significant differences between formal carers in the experimental and control group were found for approaches towards dementia (person-centredness, hope), or for whether the formal carers got to know the person with dementia and their loved one better as a result of the activity (Table 3c).

The ADQ was previously found to have an internal consistency (Cronbach’s alpha) of 0.73 for total ADQ, 0.72 for the Hope sub-scale, and 0.74 for the Person-centredness sub-scale, in a sample of formal carers in the nursing home ([36]). In this study, Cronbach’s alpha coefficient was found to be 0.54 for total ADQ, 0.45 for the Hope subscale, and 0.65 for the Person-centeredness subscale.

## 4. Discussion

This study investigated the effects of a digital Photo-Activity (experimental intervention) versus a general conversation activity (control intervention), for residents with dementia in the nursing home and their (in)formal carers. The results partially supported the first hypothesis. Compared to residents in the control group, residents in the Photo-Activity group had better positive affect (taken as an aspect of quality of life ([31]), as observed by the Independent Assessors. These findings are in line with previous studies that showed increased positive affect when residents were engaged in pleasurable activities ([68]) and greater positive affect or pleasure when the activities were person-centred, i.e., matched to the personal interests of the residents with dementia ([53]; [54]; [82]). An interesting finding from our study is that the effect on positive affect was maintained two weeks after the intervention was stopped, unlike in a previous study where significantly less pleasure was observed after the withdrawal of the intervention ([53]).

Residents who did the Photo-Activity were also observed to have significantly better mood ([10]) during the conversations with their formal carer compared to the control group. This is in line with the finding from the Photo-Activity pilot study by Theijsmeijer et al. ([77]) where a tendency for better observed mood was found in the residents who viewed the person-oriented generic photos. It also echoes the findings from the process evaluation ([74]), where Photo-Activity residents significantly more often reported that they felt positive about the conversation they had with their carer, compared to residents in the control group. Other similar results were listed in a systematic review ([73]) on psychosocial interventions that used generic photos in the intervention ([29]; [39]; [79]; [86]).

Level of dementia severity was found to influence the effects of the Photo-Activity on the residents, where the intervention seemed to work well for residents with less severe dementia (they were observed to have better mood, better stimulation level, and less need for prompting ([10]) compared to the control group), while the Photo-Activity residents with more severe dementia did worse compared to the control group residents. This finding is in line with another study where it was found that residents with less severe dementia (i.e., better cognitive functioning) showed greater pleasure during social interactions ([48]). It is possible that the Photo-Activity was complex for (some) residents with more severe dementia, affecting the way they experienced the activity. Previous research showed that taking the residents’ functioning and skill level into account, along with their personal interests increased the pleasure that residents felt from the activity, compared to when only personal interest was taken into account ([53]). However, it is also possible that the Photo-Activity did not work as well for the residents with more severe dementia due to formal carers experiencing more difficulty in holding the conversation. In the process evaluation, formal carers mentioned that they felt the Photo-Activity was not suitable for residents with more severe dementia ([74]). This perception that the Photo-Activity is ‘not for everyone’ ([74]) may have also affected the way the formal carers implemented the activity with the resident, for example, they may have entered into the activity with less of an open mind and therefore invested less energy in facilitating the activity.

The results did not support our second hypothesis. Being involved in the Photo-Activity intervention through preparation of the activity did not increase informal carers’ sense of competence. The protocol involved the informal carers in the Photo-Activity in the preparation stage (when formal carers were asked to interview informal carers regarding the personal interests of the resident with dementia). Formal carers were also encouraged to provide feedback about the Photo-Activity to the informal carers during the intervention period. However, based on the data gathered in the process evaluation ([74]), we know that informal carers did not receive a lot of updates or communication from the formal carers during the intervention period, which could explain why there was no difference between their sense of competence and that of carers in the control group who did not receive an update anyway. This was unfortunate as encouraging communication and improving relationships between formal and informal carers potentially can not only help to improve the carers outcomes but also the quality of care for residents in the nursing home ([56]). Similarly to previous research ([44]), informal carers in this study also expressed wanting more communication with the formal carers ([74]), and future work could put more emphasis on the importance of communication with family or informal carers during the training for formal carers.

The current study intended to also invite informal carers to deliver the Photo-Activity or the general conversation activity to their loved one in the nursing home to supplement and address the limited availability of formal carers, however in the first period this was not possible due to COVID-19 restrictions, and after that no carers accepted the invitation as they did not have time to do the activities with the prescribed frequency and duration for one month.

In this study, there were no significant differences between the Photo-Activity and control group after the intervention with regard to Feeling Known for the residents and informal carers. It is worth noting however that both participants in the Photo-Activity and control groups mostly gave high ratings of 7 or 8 out of a total of 10 in terms of feeling known as a person and feeling satisfied with their stay in the nursing home (for the residents), and feeling recognized as the family carer, feeling satisfied with their loved one’s stay in the nursing home, and feeling that their loved one is treated by staff in a person-centred way (for the informal carers).

Finally, the third hypothesis was also not supported by the results. The Photo-Activity did not improve formal carers’ attitudes or approach to dementia (overall, or in terms of person-centredness or hope). However, it was observed that after the intervention the formal carers’ empathy towards the residents in the Photo-Activity group was lower than that of the carers in the control group, even when baseline approach towards dementia (person-centredness, hope) of carers was taken into account (the difference in empathy was even higher for formal carers who scored high on person-centredness and hope at baseline). This was an unexpected finding because positive attitudes towards dementia have been found to result in increased empathy in formal carers ([7]; [61]). A possible explanation for this finding could be that the Photo-Activity was considered more burdensome by the formal carers compared to the control activity, which could have had a negative effect on their empathy ([27]). A consequence of the emotional labour that formal carers of people with dementia in the nursing home often face, is that detachment from the person with dementia they are caring for is necessary for them to do their work efficiently ([83]). We considered the emotional burden or emotional labour that may have been experienced during the one month intervention period in this study, however the results from our process evaluation ([74]), did not indicate that the activity led to emotional burden for the formal carers, rather it seemed more related to burden in terms of providing the intervention with their regular work. The observed decrease in empathy may be associated with experienced additional burden of learning and providing a new intervention. As formal carers’ levels of burnout were not quantitatively measured in this study, we do not know if this was the case. Previous research showed that formal carers of people with dementia generally experience low to moderate burnout levels ([20]). As mentioned above, burnout of formal carers from demanding workload could negatively affect empathy, and the effectiveness of the psychosocial intervention itself ([8]; [27]; [56]).

The Photo-activity was new to the carers in our study and additional to their regular work, and although training was provided in offering the Photo-Activity and person-centered communication strategies, this was only of a short duration and online. Further support and practical face-to-face training (as formal carers reported wanting more of in our process evaluation, [74]) for formal carers during the introduction and implementation of a psychosocial intervention could be beneficial in preventing feelings of (extra) burden, along with providing opportunities for personal reflection on their interactions with the residents ([56]; [69]; [83]).

While heavy workload and overburden are indeed barriers to successful implementation of psychosocial interventions ([56]), this explanation would contradict the results of the process evaluation where perception of time and effort investment did not differ between the Photo-Activity and control formal carers ([74]), which means that it might not have been task burden that affected empathy. There may be an alternative explanation: Empathic concern was defined as ‘other-oriented feelings of sympathy and concern for unfortunate others ([22]). For the Photo-Activity formal carers, having the digital photos to focus on together with the resident, hearing their thoughts associated with them and their personal stories, may have shifted their attention from the disabilities of the resident to their characteristics and interests as a person, while for the formal carers in the control activity, having no other conversation tool during the session, experiencing more difficulty with keeping the conversation going, may have made the resident’s current state more prominent in their minds, leading to more empathic concern.

Formal carers who did the Photo-Activity reported getting to know the residents significantly better ([74]). However, there was no difference between the Photo-Activity (experimental) and control group when formal carers were asked if they got to know the resident and the informal carer (or family member) better. An interpretation for these findings could be that while formal carers did get to learn more about the resident with dementia in the Photo-Activity, they did not learn much about the informal carer of the resident within the one-month intervention period, as there was little communication reported between them ([74]).

### 4.1. Strengths and Limitations of the Study

This study had several strengths. It aimed to examine the effects of a psychosocial intervention not just on the residents with dementia in the nursing home, but also on their formal and informal carers. In the current literature, it seems that interventions generally focus on either the person with dementia and their informal carer ([27]; [75]) or the person with dementia and the formal carers or nursing home staff ([55]; [64]), but rarely on all three. Multiple nursing homes and wards were also involved, which helps to prevent selection bias and promotes generalizability of the results. The study was designed as an RCT, to further explore the effects of the pilot study ([77]). This addresses the finding from the systematic review which saw very few studies that used generic photos in psychosocial interventions for nursing home residents with dementia ([73]). Data was collected from various participants at different times, which helped validate the findings on improved mood and positive affect in Photo-Activity residents. In terms of analysis, we also used both completers’ analysis and intention-to-treat analysis and were able to arrive to similar conclusions.

The current explorative RCT also had several limitations. First, due to difficulty in recruitment (because of the period of COVID-19 restrictions, and nursing homes facing staffing pressures), we did not reach the required sample size for the study to have enough power to detect between-group differences with large effect sizes. Thus, although we were able to show some significant differences between the intervention and control groups, other effects may have remained hidden. In addition, as this study is exploratory in nature, the sample size is not sufficiently large to test interaction effects, which typically require greater sample sizes than testing main effects to maintain sufficient power. Nor is it adequate for addressing the multiple comparisons problem. Nevertheless, the findings may serve as indicative trends for future, more definitive studies with larger sample sizes designed to test interaction effects while applying Bonferroni correction to reduce the likelihood of false positives.

As informal carers reported wanting more information and updates from the formal carers, future work could put more emphasis on the importance of communicating with informal or family carers in the training for formal carers. Also, although informal carers were involved in the process of preparing the Photo-Activity and got feedback on the residents reactions on it, we were unable to recruit any informal carer to deliver the activities with the residents, so we could not make any conclusions on whether providing the Photo-Activity as an informal carer had any effect on their relationship with the resident. Although in this study the formal carers were the primary intervention providers, making the intervention more flexible in terms of duration and frequency, and providing various iterations of the intervention (for example, remote adaptations) could help address the barriers faced by informal carers in terms of availability ([56]), and encourage more participation from informal carers in future research, which would be needed to get insight into the feasibility of the intervention when provided by informal carers. Future work could also focus on emphasizing the positive aspects of the intervention (for example, an opportunity to connect through a pleasurable activity) to the informal carers, as it has been suggested that highlighting positive aspects of caregiving can also be beneficial for informal carers’ self-efficacy and encourage continued involvement in the nursing home ([27]).

Furthermore, for the formal carers, including a quantitative measure of burn-out, as discussed earlier, could also help with clarifying the results regarding empathy.

Because the study had to be adapted during the COVID-19 pandemic, off-site monitoring of the study protocol had to be performed via video calls and emailing, resulting in some delays in terms of data collection. Potential measurement errors of self-reported outcomes such as social desirability and recall biases ([52]) also needs to be acknowledged.

It was also difficult to monitor whether assigned conditions of residents and formal carers were kept blinded, as the nursing homes that we worked with were designed to be small-scale. Student researchers who observed the activities were not blinded and knew about both conditions. The researchers however had tried to present both the experimental and control activities as beneficial to the resident, during the training of the formal carers and the student researchers.

It was decided to have two (clinical psychologist) student researchers trained in performing the activities with a few residents in the nursing home. This lowered the sample size for the formal carers, but due to the current known difficulties faced by most nursing homes in staffing and workload ([56]), this was agreed to be the best solution in order to increase resident sample size. One positive insight gained from conducting the investigation into effects of the digital Photo-Activity during the COVID-19 pandemic, however, was that the intervention and the app itself were easy enough to learn via off-site and online training, which makes the digital Photo-Activity intervention a good candidate for wider dissemination and up-scaling in more nursing homes.

### 4.2. Scientific and Clinical Value of the Study and Recommendations

This is the first study using an explorative RCT design showing that the Photo-Activity with generic photos compared to a general conversation activity had additional benefits in mood and positive affect during the activity and in the daily lives of the nursing home residents. Moreover the effect on positive affect was maintained at the two-week follow-up for residents of the Photo-Activity.

This study also found moderate to large effect sizes in several other outcomes of interest, but they were not statistically significant possibly due to the small sample size. It is therefore difficult to make any conclusions about these in this paper. Nevertheless, it provides a good direction to explore further in a larger sample size in future studies. In this study, it was also shown that using the SFAS to measure mood of the residents with dementia was feasible, allowing the researchers to collect data from the people with dementia themselves.

In terms of clinical value, the use of technology ([4]) in this study proved feasible. It not only contributed to being able to run the protocol remotely and during times that physical visits in the nursing home from the research team were not possible, it also confirmed that the digital tablet-based Photo-Activity was feasible to use with residents with moderate to advanced dementia ([74]), easy implementable in nursing homes by care staff and effective in terms of improving mood and positive affect of residents.

Positive mood can lead to a better quality of life for people with dementia residing in nursing homes ([42]; [43]). Feeling pleasure from an activity can also encourage more social engagement from the residents and improve the quality of their social interactions in the nursing home ([71]) and therewith their social health. Based on the findings that the Photo-Activity’s effects on some aspects of social interaction (mood, need for prompting, stimulation level) were influenced by the residents’ dementia severity, it is suggested that future research could focus on residents with less severe dementia first, as this was the sub-group that showed improvement during the Photo-Activity. It is also worth exploring further why formal carers providing the Photo-Activity had less empathy than those who had a general conversation with residents. Encouraging family or other informal carers to participate in providing the Photo-Activity in future studies is also recommended to be able to see if (co-)providing the Photo-Activity can help to improve carers’ sense of competence and can strengthen the relationships between resident, informal carer, and formal carer.

## 5. Conclusions

As the person-centred Photo-Activity led to residents’ improved mood and positive affect, it seems worth continuing its implementation in nursing homes and further investigate in a larger sample if other improvements in social interaction and quality of life will also be seen. While we cannot make definitive conclusions on whether overall social interaction was improved for residents who did the Photo-Activity, the improvement in mood and positive affect could definitely lead to improved social interactions because, in general, residents engage more with activities they find enjoyable. Residents with less severe dementia in the nursing home may benefit more from the intervention. Formal carers who did the Photo-Activity reported getting to know the resident better. At the same time, formal carers who did the Photo-Activity, especially those who had a more person-centred attitude and hope, showed less empathy than formal carers who had general conversations with the resident. This may be associated with developing another view on the resident as a person (instead of a disabled individual who needs care) due to the Photo-Activity, which needs to be investigated further. In general participants in both the Photo-Activity and control group seemed to respond positively to the questions about feeling more known by the carer. Further research is needed to conclude on whether the digital Photo-Activity can improve the informal carers’ sense of competence and can strengthen relationships between the residents with dementia, and their informal and formal carers.

## Figures and Tables

**Figure 1 behavsci-15-01008-f001:**
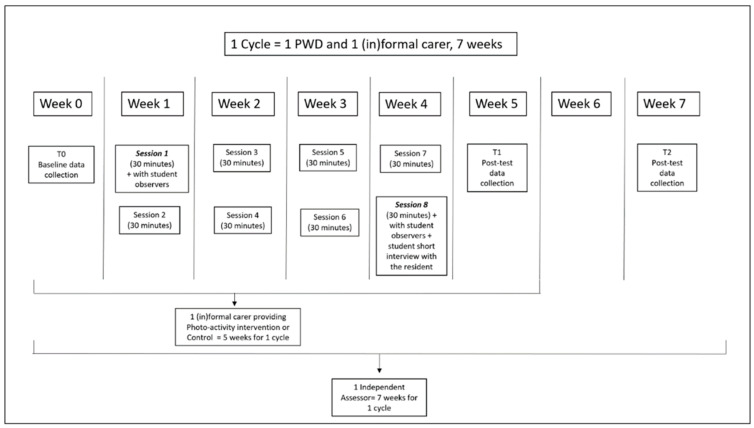
Schedule of data collection and intervention period.

**Figure 2 behavsci-15-01008-f002:**
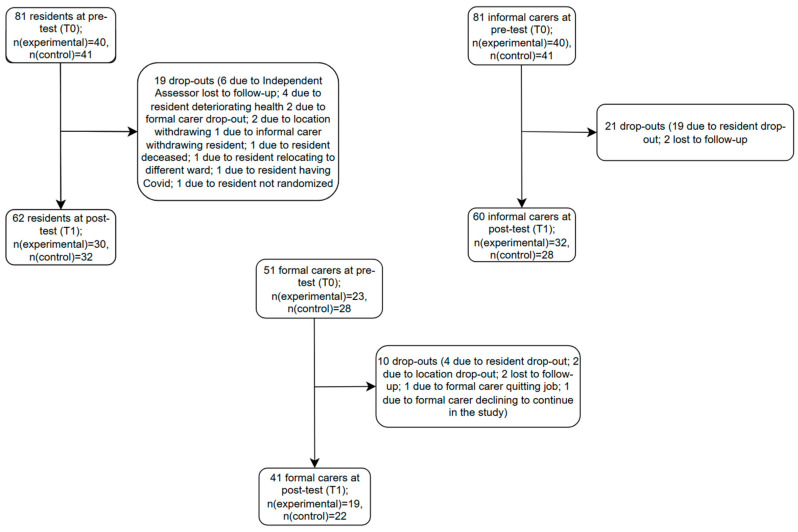
Recruitment and drop-outs of residents with dementia, informal and formal carers.

**Table 1 behavsci-15-01008-t001:** Schedule of questionnaires completed by participants.

Participant	T0	During Intervention Period	T1	T2
Research Assistant/Resident with dementia		INTERACT Observation Scale (Session 1 and 8)		
Independent Assessor	QUALIDEM NPI-Q		QUALIDEM NPI-Q	QUALIDEM NPI-Q
Resident with dementia	GDS (via ward *Coordinator*)	SFAS (1st and 4th intervention week)		
Formal Carer	TOPICS-MDS ADQ IRI		ADQ IRI	
Informal Carer	TOPICS-MDS SSCQ		SSCQ	SSCQ

Note: QUALIDEM: Quality of Life measurement scale; GDS: Global Deterioration Scale; NPI-Q: brief Neuropsychiatric Inventory Questionnaire; TOPICS-MDS: The Older Persons and Informal Caregivers Survey; ADQ: Approaches to Dementia Questionnaire; IRI: Interpersonal Reactivity Index (IRI); SSCQ: Short Sense of Competence Questionnaire (SSCQ); INTERACT: social interaction observation scale; SFAS: Smiley face assessment scale.

**Table 2 behavsci-15-01008-t002:** Background characteristics of residents with dementia, informal carers and formal carers.

		Residents				Informal Carers				Formal Carers		
Characteristics	Exp (*N* = 40)	Control (*N* = 41)	Difference Test	*p*	Exp (*N* = 40)	Control (*N* = 41)	Difference Test	*p*	Exp (*N* = 23)	Control (*N* = 28)	Difference Test	*p*
*Sex n (%)*												
Female	33 (82.5)	32 (78.0)	χ^2^ = 0.050 ^a^	0.823	9 (22.5)	11 (27.5)	χ^2^ = 0.038 ^a^	0.846	1 (4.3)	1 (3.6)	χ^2^ < 0.000 ^a,b^	1.000
*Age M (SD),* [min-max]	84.58 (7.52), [68–96]	85.18 (7.38), [62–96]	t(74) = 0.354	0.724	58.00 (1.455), [31–75]	59.00 (1.725), [27–84]	U = 619.500 Z = −1.240	0.215	50.00 (2.707), [20–62]	51.50 (2.776), [18–63]	U = 309.000 Z = −0.246	0.805
*Marital Status n (%)*												
1 = Married	10 (25)	8 (19.5)	χ^2^ = 6.377 ^b^	0.173	N/A ^d^	N/A			N/A	N/A		
2 = Unmarried, no partner	4 (10)	0										
3 = Divorced	3 (7.5)	5 (12.2)										
4 = Widow/widower/partner deceased	21 (52.5)	25 (61.0)										
*Place of birth n (%)*												
Netherlands	37 (92.5)	36 (87.8)	χ^2^ = 0.347 ^b^	0.556	N/A	N/A			N/A	N/A		
Other	1 (2.5)	2 (4.9)										
*Education level n (%)*												
Elementary	8 (20)	10 (24.4)	U = 657.500 Z = −0.747	0.455	0	4 (9.8)	U = 710.500 Z = −0.593	0.593	0	0	U = 292.000 Z = −0.950	0.342
Lower Education	26 (65)	17 (41.5)			20 (50)	13 (31.7)			21 (91.3)	23 (82.1)		
Advanced Education	1 (2.5)	6 (14.6)			5 (12.5)	5 (12.2)			1 (4.3)	2 (7.1)		
University/Higher Education	3 (7.5)	5 (12.2)			13 (32.5)	18 (43.9)			1 (4.3)	3 (10.7)		
*Type of dementia n (%)*												
Alzheimer’s Disease	23 (57.5)	32 (78)	χ^2^ = 3.036 ^a^	0.081	N/A	N/A			N/A	N/A		
Vascular dementia	7 (17.5)	2 (4.9)										
Other	10 (25.0)	7 (17.1)										
*Severity of dementia n (%)*												
Mild to moderate (GDS ^c^ 4 and 5)	27 (67.5)	27 (65.9)	U = 806.500 Z = −0.156	0.876	N/A	N/A			N/A	N/A		
Severe (GDS ^c^ 6)	13 (32.5)	14 (34.1)										
*Years in nursing home*												
n, Mdn (SE), [min–max]	n = 38, 1.050 (0.2200), [0.2–7.0]	n = 37, 1.3 (0.3132), [0.2–10.0]	U = 632.500 Z = −0.748	0.454								
*Relation to Resident n (%)*												
Husband/Wife/Partner	N/A	N/A			6 (15)	8 (19.5)	χ^2^ = 1.051 ^b^	0.789	N/A	N/A		
Son/Daughter					4 (10)	5 (12.2)						
Daughter/son-in-law					24 (60)	24 (58.5)						
Other					4 (10)	2 (4.9)						
*Paid work*												
Yes n (%)	N/A	N/A			27 (67.5)	23 (56.1)	χ^2^ = 1.555	0.212	N/A	N/A		
*Hours spent caring for resident in the last week n (%)*												
Low (0–3 h)	N/A	N/A			20 (50)	20 (48.8)	U = 741.500 Z = 0.240	0.81	N/A	N/A		
High (4 or more hours)					19 (47.5)	17 (41.5)						
*Work Function n (%)*												
Carer	N/A	N/A			N/A	N/A			7 (30.4)	6 (21.4)	χ^2^ = 1.931	0.587
Nurse									1 (4.3)	0		
Activity therapist									4 (17.4)	6 (21.4)		
Other									11 (47.8)	16 (57.1)		
*Years of work experience in psychogeriatrics* Mdn (SE), [min-max]	N/A	N/A			N/A	N/A			4.000 (0.9630), [1.0–16.0]	5.500 (2.1191), [1.0–40.0]	U = 258.500 Z = −1.212	0.226

^a^ with Yate’s continuity correction. ^b^ includes cells that have expected count less than 5. ^c^ Global Deterioration Scale ([65]). ^d^ Not applicable.

**Table 3 behavsci-15-01008-t003:** (**a**) ANCOVA main effects analyses for outcomes of residents with dementia (ranges are shown per scale or sub-scale, with the positive or desirable score underlined). (**b**) ANCOVA main effects analyses for outcomes of informal carers (ranges are shown per scale or sub-scale, with the positive or desirable score underlined). (**c**) ANCOVA main effects analyses for outcomes of formal carers (ranges are shown per scale or sub-scale, with the positive or desirable score underlined).

(a)												
Resident	Pre-Test (T0)				Post-Test (T1)				ANCOVA			Effect Size
	Experimental (n = 38)		Control (n = 39)		Experimental (n = 31)		Control (n = 33)		Experimental adjM (SE)	Control adjM (SE)	*p*	Partial Eta^2^
*INTERACT*	M	(SD)	M	(SD)	M	(SD)	M	(SD)				
Mood (range: 0–24)	19.63	(4.01)	17.44	(4.69)	19.39	3.12	16.88	3.90	18.97 (0.56)	17.28 (0.54)	0.037 *****	0.070
Speech (range: 0–20)	16.17	(4.03)	14.94	(4.16)	15.45	5.14	14.36	4.47	15.12 (0.69)	14.68 (0.67)	0.645	0.003
Relating to person (range: 0–24)	19.06	(3.38)	17.97	(2.40)	18.10	3.29	17.82	2.83	17.83 (0.50)	18.07 (0.48)	0.733	0.002
Relating to environment (range: 0–20)	15.83	(3.45)	14.58	(4.16)	14.48	3.98	15.24	4.17	14.49 (0.74)	15.25 (0.72)	0.473	0.008
Need for prompting (range: 0–4)	2.86	(1.14)	2.39	(1.29)	2.39	1.38	2.36	1.19	2.31 (0.22)	2.43 (0.21)	0.685	0.003
Stimulation level (range: 0–20)	17.63	(2.97)	16.11	(3.50)	17.58	2.90	16.24	3.11	17.32 (0.52)	16.49 (0.50)	0.261	0.021
Negative interactions (range: 0–24)	23.63	(0.84)	22.64	(2.00)	23.74	0.514	22.67	1.98	23.56 (0.25)	22.84 (0.25)	0.053	0.060
*NPI-Q* (range: 0–30)	(n = 35)		(n = 36)		(n = 30)		(n = 32)					
Total Severity Score	3.00	(3.24)	3.97	(4.65)	3.17	(4.56)	2.97	(3.62)	3.42 (0.55)	2.73 (0.53)	0.377	0.013
*SFAS* (range: 1–5)	(n = 35)		(n = 35)		(n = 35)		(n = 35)					
Mean SFAS Score of weeks 1 and 4	3.57	(0.59)	3.38	(0.64)	3.92	(0.64)	3.83	(0.69)	3.87 (0.10)	3.87 (0.10)	0.996	0.000
*Feeling Known* (range: 0–10)	(n = 32)		(n = 32)		(n = 25)		(n = 23)					
Known as a person in the nursing home	7.06	(1.98)	6.69	(2.67)	7.88	(1.20)	7.26	(2.05)	7.75 (0.27)	740 (0.28)	0.367	0.018
	(n = 34)		(n = 34)		(n = 25)		(n = 23)					
Satisfied with staying in the nursing home	7.74	(1.64)	7.03	(2.66)	8.00	(1.54)	7.58	(1.92)	7.75 (0.28)	7.83 (0.28)	0.845	0.001
*QUALIDEM*	(n = 38)		(n = 39)		(n = 30)		(n = 32)					
Care Relationship (range: 0–21)	15.39	(4.27)	14.46	(4.47)	15.90	(4.08)	14.91	(5.01)	15.50 (0.56)	15.28 (0.54)	0.774	0.001
Positive Affect (range: 0–18)	14.21	(3.81)	14.69	(3.12)	15.20	(3.69)	13.69	(3.85)	15.12 (0.42)	13.76 (0.41)	**0.025 ***	0.082
Negative Affect (range: 0–9)	5.37	(2.75)	5.44	(2.25)	6.10	(2.52)	5.66	(1.89)	6.03 (0.25)	5.72 (0.24)	0.365	0.014
Restless Tense-Behaviour (range: 0–9)	5.29	(2.45)	4.90	(2.72)	5.97	(2.83)	5.03	(2.94)	5.71 (0.37)	5.27 (0.36)	0.399	0.012
Positive Self-Esteem (range: 0–9)	6.42	(2.31)	6.33	(1.97)	7.10	(2.11)	6.75	(2.02)	7.01 (0.22)	6.83 (0.21)	0.558	0.006
Social Relationships (range: 0–18)	10.55	(2.39)	10.49	(2.76)	11.50	(2.71)	10.50	(2.65)	11.42 (0.41)	10.57 (0.39)	0.137	0.037
Social Isolation (range: 0–9)	6.87	(2.16)	6.08	(2.24)	7.30	(1.71)	6.34	(1.99)	7.07 (0.24)	6.56 (0.23)	0.138	0.037
Feeling at Home (range: 0–12)	8.66	(2.87)	8.59	(3.39)	8.97	(3.49)	8.50	(3.23)	8.79 (0.36)	8.67 (0.34)	0.806	0.001
Have Something to Do (range: 0–6)	3.18	(1.67)	2.97	(1.71)	3.47	(1.74)	3.13	(1.60)	3.40 (0.23)	3.18 (0.22)	0.488	0.008
(b)												
Informal Carer	Pre-test (T0)				Post-test (T1)				ANCOVA			
	Experimental (n = 37)		Control (n = 37)		Experimental (n = 32)		Control (n = 28)		Experimental	Control	*p*	Effect Size Partial Eta^2^
*SSCQ* (range: 7–35)	M	(SD)	M	(SD)	M	(SD)	M	(SD)	adjM (SE)	adjM (SE)		
Sense of Competence	28.73	(5.21)	26.97	(5.22)	28.34	(3.81)	28.43	(4.51)	27.88 (0.57)	28.96 (0.61)	0.208	0.028
*Feeling Known* (range: 1–10)	(n = 37)		(n = 37)		(n = 32)		(n = 28)					
Feel seen as the family carer in the nursing home	8.32	(1.6)	8.51	(1.30)	8.47	(1.61)	8.39	(1.55)	8.57 (0.14)	8.8 (0.15)	0.159	0.034
Satisfied with the stay of your loved one in the nursing home	8.65	(1.18)	8.57	(1.35)	8.72	(1.20)	8.54	(1.84)	8.66 (0.13)	8.60 (0.14)	0.729	0.002
My loved one with dementia is known in the nursing home	8.78	(1.25)	8.57	(1.32)	8.72	(1.14)	8.39	(1.69)	8.68 (0.16)	8.44 (0.17)	0.313	0.018
(c)												
Formal Carer	Pre-Test (T0)				Post-Test (T1)				ANCOVA			
	Experimental (n = 23)		Control (n = 28)		Experimental (n = 19)		Control (n = 22)		Experimental adjM (SE)	Control adjM (SE)	*p*	Effect Size Partial Eta^2^
*IRI*	M	(SD)	M	(SD)	M	(SD)	M	(SD)				
Global Empathy (range: 0–112)	64.04	(8.46)	59.86	(9.34)	59.58	(6.45)	62.36	(8.70)	58.07 (1.37)	63.66 (1.27)	0.006 *	0.185
*Subscales (range: 0–28)*												
Perspective Taking	19.09	(3.04)	18.57	(3.10)	18.37	(2.54)	18.95	(2.77)	18.09 (0.47)	19.19 (0.44)	0.096	0.071
Empathic Concern	19.09	(2.80)	17.36	(3.42)	17.95	(2.51)	18.73	(3.07)	17.32 (0.50)	19.27 (0.47)	0.009 *	0.167
Fantasy	14.96	(4.01)	13.54	(4.22)	12.74	(3.51)	14.45	(3.20)	12.27 (0.60)	14.86 (0.56)	0.003 *	0.203
Personal Distress	10.91	(3.34)	10.39	(3.56)	10.53	(4.21)	10.28	(4.12)	10.41 (0.63)	10.33 (0.59)	0.932	0.000
*ADQ*	(n = 23)		(n = 28)		(n = 19)		(n = 22)					
*Total ADQ* (19–95)	77.17	(4.22)	74.36	(4.35)	75.37	(4.35)	74.86	(5.12)	74.43 (0.99)	75.68 (0.91)	0.372	0.021
Person-Centered (score range: 11–55)	49.17	(3.73)	47.29	(3.74)	48.37	(3.32)	47.73	(3.82)	47.70 (0.66)	48.31 (0.61)	0.507	0.012
Hope (score range: 8–40)	27.91	(2.37)	26.86	(2.32)	27.21	(2.44)	26.95	(2.54)	27.09 (0.53)	27.06 (0.49)	0.961	0.000
					(n = 19)		(n = 24)		Difference Test			
*Feeling Known* (range: 1–10) Got to know the person with dementia and their loved one better	n/a ^a^	n/a	n/a	n/a	7.00	(0.39)	7.00	(0.30)	U = 237.000 Z = 0.228		0.820	0.035
*Feeling Known* n(%) ^a^ (range: 1–10) Got to know the person with dementia *(only)* better Yes Much Better Yes a Little Better No, Not Better	n/a	n/a	n/a	n/a	(n = 23) 9 (39.1) 11 (47.8) 2 (8.7)	(n = 27) 4 (14.8) 17 (63.0) 6 (22.2)	U = 390.500 Z = 2.114		0.035 *	0.299		

* Significance set to *p* < 0.05. ^a^ Not applicable.

**Table 4 behavsci-15-01008-t004:** ANCOVA Interaction effects of Group and Dementia Severity on INTERACT subscales.

		Pre-Test (T0)				Post-Test (T1)				ANCOVA				
		Experimental (n = 35)		Control (n = 36)		Experimental (n = 31)		Control (n = 33)		Experimental adjM (SE)	Control adjM (SE)	Interaction	*p*	Effect Size
*INTERACT*	GDS	M	(SD)	M	(SD)	M	(SD)	M	(SD)					Partial Eta^2^
Mood (range: 0–24)	Low	19.59	(4.33)	17.57	(4.84)	20.15	(2.46)	16.19	(3.79)	19.66 (0.68)	16.59 (0.66)	GroupxGDS	0.017 *	0.092
	High	19.69	(3.57)	17.23	(4.60)	18.00	(3.80)	18.08	(3.97)	17.71 (0.90)	18.46 (0.87)			
Speech (range: 0–20)	Low	16.27	(3.67)	15.30	(4.07)	17.10	(3.52)	14.76	(3.65)	16.53 (0.82)	14.84 (0.79)	GroupxGDS	0.075	0.053
	High	16.00	(4.73)	14.31	(4.40)	12.45	(6.36)	13.67	(5.76)	12.60 (1.09)	14.34 (1.05)			
Relating to Person (range: 0–24)	Low	19.59	(2.94)	17.61	(2.68)	18.75	(2.77)	17.62	(3.29)	18.16 (0.64)	18.06 (0.61)	GroupxGDS	0.552	0.006
	High	18.15	(3.98)	18.62	(1.71)	16.91	(3.94)	18.17	(1.85)	17.26 (0.84)	18.05 (0.80)			
Relating to Environment	Low	16.00	(3.64)	14.09	(4.30)	14.85	(4.18)	15.57	(4.20)	14.85 (0.94)	15.57 (0.92)	GroupxGDS	0.954	0.000
(range: 0–20)	High	15.54	(3.23)	15.46	(3.93)	13.82	(3.68)	14.67	(4.22)	13.82 (1.25)	14.67 (1.20)			
Need for Prompting	Low	2.77	(1.11)	2.22	(1.28)	2.90	(1.17)	2.29	(1.27)	2.84 (0.25)	2.45 (0.25)	GroupxGDS	0.013 *	0.099
(range: 0–4)	High	3.00	(1.23)	2.69	(1.32)	1.45	(1.29)	2.50	(1.09)	1.33 (0.33)	2.42 (0.32)			
Stimulation level	Low	17.64	(2.94)	15.96	(3.62)	18.45	(1.91)	15.62	(3.43)	18.17 (0.62)	15.95 (0.61)	GroupxGDS	0.011 *	0.106
(range: 0–20)	High	17.62	(3.15)	16.38	(3.40)	16.00	(3.74)	17.33	(2.19)	15.83 (0.83)	17.38 (0.79)			
Negative Interactions	Low	23.68	(0.65)	22.87	(1.91)	23.75	(0.44)	22.38	(2.22)	23.52 (0.31)	22.48 (0.30)	GroupxGDS	0.166	0.032
(range: 0–24)	High	23.54	(1.13)	22.23	(2.17)	23.73	(0.65)	23.17	(1.40)	23.57 (0.41)	23.52 (0.40)			

* Significance set to *p* < 0.05.

**Table 5 behavsci-15-01008-t005:** Mixed model for repeated measures (MMRM) results for outcomes across T0, T1 and T2. (ranges are shown per scale or sub-scale, with the positive or desirable score underlined).

		Means		Effect Estimate (GroupxTime Interaction)		*p*-Value	Effect Size *(Cohen’s d)*
Qualidem Subscales (Range)	**T0**	**T1**	**T2**				
*Care Relationships* (0–21)							
Experimental	15.96	16.34	16.09	GroupxT1	0.27	0.712	0.06
Control	15.96	16.07	15.68	GroupxT2	0.41	0.611	0.09
*Positive Affect* (0–18)							
Experimental	15.07	15.40	15.28	GroupxT1	1.31	0.028 *	0.38
Control	15.07	14.09	13.93	GroupxT2	1.34	0.042 *	0.39
*Negative Affect* (0–9)							
Experimental	6.15	6.67	6.38	GroupxT1	0.29	0.398	0.12
Control	6.15	6.38	6.76	GroupxT2	−0.38	0.371	−0.15
*Restless Tense Behavior* (0–9)							
Experimental	5.02	5.47	5.01	GroupxT1	0.48	0.341	0.19
Control	5.02	4.99	5.29	GroupxT2	−0.28	0.516	−0.11
*Positive Self-Esteem* (0–9)							
Experimental	6.93	7.39	7.24	GroupxT1	0.12	0.693	0.06
Control	6.93	7.27	7.16	GroupxT2	0.09	0.821	0.04
*Social Relationships* (0–18)							
Experimental	10.50	11.13	10.31	GroupxT1	0.83	0.139	0.32
Control	10.50	10.30	9.78	GroupxT2	0.53	0.246	0.21
*Social Isolation* (0–9)							
Experimental	5.78	6.29	6.16	GroupxT1	0.53	0.091	0.24
Control	5.78	5.76	5.70	GroupxT2	0.46	0.300	0.20
*Feeling at Home* (0–12)							
Experimental	8.59	8.45	8.64	GroupxT1	0.06	0.906	0.02
Control	8.59	8.40	8.39	GroupxT2	0.25	0.612	0.08
*Having Something to Do* (0–6)							
Experimental	2.70	2.90	3.04	GroupxT1	0.17	0.572	0.10
Control	2.70	2.73	2.54	GroupxT2	0.50	0.131	0.30
*NPI-Q* (10) total (range: 0–30)							
Experimental	4.22	4.15	4.13	GroupxT1	0.71	0.341	0.18
Control	4.22	3.44	3.51	GroupxT2	0.62	0.495	0.15
*Short Sense of Competence total* (range: 7–35)							
Experimental	29.51	29.28	29.62	GroupxT1	−1.00	0.227	−0.19
Control	29.51	30.28	30.60	GroupxT2	−0.98	0.251	−0.19
* Questions for Resident *							
*Regarding feeling known as a person (range: 1–10)*							
Experimental	7.69	8.27	8.05	GroupxT1	0.50	0.213	0.21
Control	7.69	7.77	7.83	GroupxT2	0.22	0.611	0.09
*Regarding feeling satisfied with nursing home care* *(range: 1–10)*							
Experimental	7.51	7.70	7.30	GroupxT1	−0.04	0.918	−0.02
Control	7.51	7.74	7.23	GroupxT2	0.07	0.840	0.03
* Questions for Informal Carer *							
*Regarding feeling known as the informal carer* *(range: 1–10)*							
Experimental	8.60	8.74	8.59	GroupxT1	0.28	0.168	0.19
Control	8.60	8.46	8.51	GroupxT2	0.07	0.731	0.05
*Regarding feeling satisifed with loved one’s stay in nursing home* *(range: 1–10)*							
Experimental	8.69	8.66	8.57	GroupxT1	0.08	0.672	0.06
Control	8.69	8.58	8.37	GroupxT2	0.19	0.464	0.15
*Regarding feeling that loved one is known in nursing home* *(range: 1–10)*							
Experimental	8.79	8.75	8.64	GroupxT1	0.25	0.276	0.19
Control	8.79	8.51	8.44	GroupxT2	0.20	0.380	0.16

* Significance set to *p* < 0.05.

**Table 6 behavsci-15-01008-t006:** ANCOVA interaction effects between baseline Approach towards dementia scores (ADQ) and Global empathy (IRI; ranges are shown per scale or sub-scale, with the positive or desirable score underlined).

		Pre-Test (T0)				Post-Test (T1)				ANCOVA				Effect Size
		Exp (n = 23)		Control (n = 28)		Experimental (n = 19)		Control (n = 22)		Experimental adjM (SE)	Control adjM (SE)		*p*	Partial Eta
*Global* *Empathy IRI ^a^* (range: 0–112)	*Baseline Total ADQ ^b^*	M	(SD)	M	(SD)	M	(SD)	M	(SD)			Group		
	Low	67.14	(9.03)	57.79	(7.15)	64.83	(4.31)	59.73	(7.32)	61.67 (2.41)	62.13 (1.55)	Total Baseline ADQ ^b^	0.851	0.001
	High	62.69	(8.11)	64.22	(12.15)	57.15	(5.87)	68.00	(9.24)	56.76 (1.57)	66.32 (2.16)	Group xTotal Baseline ADQ ^b^	0.032 *	0.121

^a^ IRI = Interpersonal Reactivity Index. ^b^ ADQ = Approaches towards Dementia Questionnaire. * Significance set to *p* < 0.05.

## Data Availability

The original contributions presented in this study are included in the article. Further inquiries can be directed to the corresponding author.

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
