# Peer review of "Effects of a Digital, Person-Centered, Photo-Activity Intervention on the Social Interaction of Nursing Home Residents with Dementia, Their Informal Carers and Formal Carers: An Explorative Randomized Controlled Trial"

_behavsci, 2025, doi:10.3390/bs15081008_

Round 1
Reviewer 1 Report
Comments and Suggestions for Authors
The article “Effects of a digital, person-centered, Photo-Activity intervention on the social interaction of nursing home residents with dementia, their informal carers and formal carers: an explorative randomized controlled trial” represents a significant contribution to the scientific community by proposing and testing a person-centered digital intervention targeting nursing home residents with dementia, involving both formal and informal caregivers.
The article appropriately follows the expected structure of a scientific paper. The introduction clearly presents the problem; the literature review provides relevant theoretical and empirical justifications for the proposed intervention; the methodology aligns well with the study's objectives; and the results are presented with the support of tables and appropriate statistical analyses. The discussion section is connected to the research questions and attempts to critically interpret the findings.
However, despite being a well-written scientific article, there are aspects that could be improved, as outlined below:
- Although the research questions are well formulated, explicit research hypotheses are not presented. This omission weakens the connection between the theoretical framework and the statistical analysis. It is recommended that the authors formulate operational hypotheses at the end of the introduction, for example:
“Residents participating in the Photo-Activity will show significant improvements in mood and social interaction compared to the control group.”
- While positive effects are observed on the mood and positive affect of residents (pp. 13–14), the results do not indicate significant improvements for informal caregivers (p. 14), nor in terms of empathy among formal caregivers (p. 15), which contradicts the expectations outlined:
“No significant differences in sense of competence (SSCQ)… or feeling recognized…” (p. 15)
“The control group scored higher in global empathy compared to the experimental group…” (p. 15)
It is proposed that these limitations be discussed more thoroughly in the discussion section, offering possible explanations such as insufficient training, emotional overload of the caregivers, or challenges in implementing the intervention within the institutional context.
- The fact that no informal caregiver carried out the activity partially undermines the study's initial objectives:
“…there were no informal carers who wanted to participate in the study to deliver either of the activities…” (p. 13)
Therefore, it is recommended that the authors explicitly acknowledge this limitation and propose, in future studies, more flexible strategies to engage informal caregivers, such as shorter sessions, remote adaptations, or hybrid formats compatible with caregivers’ availability and well-being.
- The decrease in empathy levels (as measured by the IRI) among formal caregivers in the experimental group is a result that, in our view, contradicts the central aim of the study:
“The control group scored higher in global empathy… [and] on the sub-scales fantasy and empathic concern.” (p. 15)
This finding warrants critical reflection. The authors are encouraged to consider the possibility that continuous exposure to the digital activity, without adequate emotional support, may have led to empathic fatigue or emotional exhaustion. It is therefore recommended to reflect on the need for psychological support or continuous supervision for professionals involved in similar interventions.
From a formal perspective, the article is well written, with clear language and a coherent structure. The tables, although extensive, are relevant and well organized. The use of mixed-model regression (MMRM, Table 5) effectively complements the ANCOVA analysis, adding robustness to the interpretation of results.
Although the authors cited are well grounded, some references are excessively outdated – for example, Kitwood & Bredin (1992), Kaufer (2000), Davis (1980) – which weakens the theoretical framework’s currency. It is recommended to complement the literature review with more recent studies (2020–2024), to better reflect the current state of the art in the fields of dementia, digital intervention, and person-centered care.
In sum, this is a rigorous and innovative study with relevant contributions to research and clinical practice in dementia care contexts. However, the limitations identified – particularly the absence of explicit hypotheses, the non-participation of informal caregivers, and the unexpected effects on formal caregivers' empathy levels – should be properly addressed in a revised version of the manuscript. Therefore, publication is recommended, provided that the revisions indicated above are considered.
Reviewer 2 Report
Comments and Suggestions for Authors
Thank you for the opportunity to review this article.
The article explored the effects of a social interaction intervention on nursing home residents and caregivers’ well-being.
The study employed an exploratory RCT. It also used validated scales. The topic is novel and very important to clinical practice and health policy. Overall, the paper is well written.
The paper can be improved in the following ways:
1) Measure: Please provide psychometric properties of the scales, e.g., Cronbach’s alpha, in your sample.
Please clarify who translated the scales and whether the translated versions were validated in your population of interest.
2) Outcome: By the definition of a RCT, there needs to be a primary outcome. Other outcomes should be treated as secondary.
Please address multiple comparison problem to avoid false positive results.
3) Discussion: The limitations section should acknowledge potential measurement errors of self-reported outcomes, such as social desirability and recall biases.
Please specifically discuss that testing the interaction effect would require a larger sample size than the one calculated from your power analysis, and thus, the results should be treated as exploratory.
4) Interpretation: When interpreting the “surprising” findings about caregivers, you could consider referencing the work on caregiver burden:
https://link.springer.com/article/10.1007/s11136-021-02782-9
https://pubmed.ncbi.nlm.nih.gov/24618967/
5) Figure: The resolution of figures could be improved, perhaps with svg vector formats.
6) Registration: You mentioned registration of your trial, but please clarify whether this is a pre-registration.
Best luck moving forward!
